# First insights into the microbiome of Tunisian *Hyalomma* ticks gained through next-generation sequencing with a special focus on *H. scupense*

**Hayet Benyedem**[1,2], **Abdelmalek Lekired**[3], **Moez Mhadhbi**[1], **Mokhtar Dhibi**[1], **Rihab Romdhane**[1], **Soufiene Chaari**[4], **Mourad Rekik**[5], **Hadda-Imene Ouzari**[3], **Tarek Hajji**[6], **Mohamed Aziz Darghouth**[1]*

**1** Laboratoire de Parasitologie, Institution de la Recherche et de l'Enseignement Supérieur Agricoles and Univ. Manouba, École Nationale de Médecine Vétérinaire de Sidi Thabet, Sidi Thabet, Tunisia, **2** Faculty of Sciences of Tunis, University of Tunis El Manar, Tunis, Tunisia, **3** Faculté des Sciences de Tunis, Laboratoire des Microorganismes et Biomolécules Actives (LR03ES03), Université Tunis El Manar, Tunis, Tunisia, **4** Laboratoire pharmaceutique vétérinaire MEDIVET, Soliman, Tunisia, **5** International Centre for Agricultural Research in the Dry Areas (ICARDA), Tunis, Tunisia, **6** Higher Institute of Biotechnology—Sidi Thabet, Laboratory of Biotechnology and Valorization of Bio-Geo-Resources (LR11ES31), Univ. Manouba, Ariana, Tunisia

* darghouth@iresa.tn

**Data Availability Statement:** All relevant data are within the paper and its Supporting Information files. Please, note the following data availability

## Abstract

Ticks are one of the most important vectors of several pathogens affecting humans and animals. In addition to pathogens, ticks carry diverse microbiota of symbiotic and commensal microorganisms. In this study, we have investigated the first Tunisian insight into the microbial composition of the most dominant *Hyalomma* species infesting Tunisian cattle and explored the relative contribution of tick sex, life stage, and species to the diversity, richness and bacterial species of tick microbiome. In this regard, next generation sequencing for the 16S rRNA (V3-V4 region) of tick bacterial microbiota and metagenomic analysis were established. The analysis of the bacterial diversity reveals that *H. marginatum* and *H. excavatum* have greater diversity than *H. scupense*. Furthermore, microbial diversity and composition vary according to the tick's life stage and sex in the specific case of *H. scupense*. The endosymbionts *Francisella*, *Midichloria mitochondri*, and *Rickettsia* were shown to be the most prevalent in *Hyalomma* spp. *Rickettsia*, *Francisella*, *Ehrlichia*, and *Erwinia* are the most common zoonotic bacteria found in *Hyalomma* ticks. Accordingly, *Hyalomma* ticks could represent potential vectors for these zoonotic bacterial agents.

## Introduction

Ticks are strict hematophagous ectoparasites of vertebrate animals and humans. Distributed all over the world, they are implicated in the transmission of several pathogens (viruses, bacteria, protozoa, and helminths) of medical and veterinary importance. Accordingly, ticks are

specification and links Accession Sample Name SPUID Organism Tax ID SAMN24296740 S19 S19 tick metagenome 1595979 SAMN24296741 S20 S20 tick metagenome 1595979 SAMN24296742 S21 S21 tick metagenome 1595979 SAMN24296743 S22 S22 tick metagenome 1595979 SAMN24296744 S23 S23 tick metagenome 1595979 SAMN24296745 S24 S24 tick metagenome 1595979 https://www.ncbi.nlm.nih.gov/biosample/24296740 https://www.ncbi.nlm.nih.gov/biosample/24296741 https://www.ncbi.nlm.nih.gov/biosample/24296742 https://www.ncbi.nlm.nih.gov/biosample/24296743 https://www.ncbi.nlm.nih.gov/biosample/24296744 https://www.ncbi.nlm.nih.gov/biosample/24296745.

**Funding:** This study was financially supported by the project of the CGIAR Research Program on Livestock (CRP Livestock), and the "Laboratory of Epidemiology of Enzootic Infections in Herbivores in Tunisia: Application to Control" (LR16AGR01) funded by the Ministry of Higher Education and Scientific Research, Tunisia. This study is also carried out under the MOBIDOC scheme for the student allowance, funded by the Ministry of Higher Education and Scientific Research through the PromEssE project and managed by the ANPR (Agence Nationale de Promotion de la Recherche Scientifique).

**Competing interests:** The authors have declared that no competing interests exist.

**Abbreviations:** rRNA, Ribosomal ribonucleic acid; 16S, Gene coding for the small ribosomal 16 RNA gene; 12S, gene coding for the small ribosomal 12 RNA gene; V3-V4, the hypervariable regions of the 16S rRNA gene; OTU, Operational Taxonomic Unit; RPA, Relative Percent Abundance; NGS, Next Generation Sequencing; *M. mitochondrii*, *Midichloria mitochondrii*.

considered to be the second world wild vectors of human diseases behind mosquitos, but they are the first vectors of pathogen diseases in domestic and wild animals [1, 2].

Livestock farming is one of the main agricultural sectors in North African countries and Tunisia [3]. Ticks are widespread in Tunisia over different geographical regions, *Hyalomma* spp are the most prevalent ticks infesting cattle [4, 5].

Tick infestation represents a major threat in cattle breeding due to their capacity to transmit three major tick-borne pathogens (TBPs) and to induce different diseases specifically anaplasmosis, babesiosis and theileriosis [4, 6, 7]. Tropical theileriosis, which is caused by the protozoan *Theileria annulata* and transmitted by *Hyalomma scupense*, is the major tick-borne disease affecting North African and Tunisian cattle [8–10].

Besides pathogens, ticks also harbor a large variety of symbiotic and commensal microorganisms [11]. Symbiotic bacteria play important roles in tick survival particularly via the synthesis of some essential vitamins and cofactors, especially vitamins of group B, which are lacking in the highly specialized hematophagous tick diet [12–14]. These bacteria are implicated in tick adaptation to environmental stress, they are also essential for tick's reproductive fitness [11, 12]. The symbiotic bacteria can also influence the colonization, maintenance and transmission of pathogens [15, 16]. Ticks carry at least 10 different genera of maternally inherited endosymbiotic bacteria [12, 13]. Three of these endosymbionts namely *Coxiella*, *Francisella* and *M. mitochondrii* are specific to ticks. *Coxiella* is the most frequent endosymbiont which presents in two-thirds of tick species [12, 17]. *M. mitochondrii* which infect the mitochondria has been reported in some tick species, while *Francisella* has only been detected in a few tick species [18]. Indeed, *Francisella*, *Coxiella* and *Rickettsia* endosymbionts are obligate nutritional symbiotic bacteria that have conserved all the major genes encoding for vitamins B synthesis despite their highly small genome [19–21]. For instance, *Coxiella* endosymbiont carries the genes encoding for vitamins B7 (biotin), B2 (riboflavin), B9 (folic acid), and their cofactors [22, 23]. In the *Rickettsia* endosymbionts genome, Hunter's team confirmed the presence of all genes encoding for the biosynthesis of folic acid [21]. Also, *Francisella* endosymbiont genome includes all the genetic pathways for the synthesis of folic acid, biotin, and riboflavin [16, 24].

The importance of B vitamins in tick survival and development has been established by Duron et al. in 2018 with *Francisella* in the soft tick *Ornithodoros moubata* [13]. An experimental elimination of the symbiont using antibiotic treatment was followed by a reduced tick survival and the appearance of some physical abnormalities, interestingly, these effects were fully restored with a supplement of vitamin B. Similar experiments with *Coxiella* in *Amblyomma americanum*, *Haemaphysalis longicornis*, and *Rhipicephalus microplus* resulted in lower reproductive fitness and prevented even the development into adults in *R. microplus*. While, being one of the maternally transmitted endosymbiotic, the role of *M. mitochondrii* in their host biology has not been well established. However, some studies have suggested their role in molting and tick blood meal metabolism following the increase of this symbiont after feeding [25, 26].

Taken altogether the above general data are highlighting the important role of obligate symbionts in tick biology and reproduction, opening a potential window of opportunities for developing prototypes of innovative and environmentally friendly control measures targeting these symbionts.

In North Africa and Tunisia, the majority of studies on livestock and zoonotic tick-borne pathogens have focused essentially on pathogens and diseases identification, and their epidemiology and control using conventional approaches. However, almost there is no data on microbiota and endosymbionts of the most dominant and important Tunisian tick species of the *Hyalomma* genus. In this context, our work intends to provide the first comprehensive

study carried out on major *Hyalomma* species in Tunisia and more generally in North Africa, for describing, firstly, the microbial diversity and richness of tick microbiota in *Hyalomma* ticks infecting livestock in different bioclimatic and geographic regions in Tunisia, and establishing secondly, the prevalence of their endosymbiotic bacteria. We aim in the present work to focus more, and for the first time to our knowledge, on the symbionts associated with the different life stages of the monotropic *H. scupense* cattle tick, the vector of tropical theileriosis, and to carry out a comparative study with the adult stages of other dominant cattle *Hyalomma* tick species in Tunisia.

## Material and methods

### Study area and tick sampling

A total of 776 adult ticks were sampled during June, July and August 2020. Adult ticks were collected from cattle in 6 different bioclimatic stages and 9 Tunisian administrative governorates to explore the whole microbial diversity of Tunisian ticks and to investigate their endosymbionts. For this purpose, cattle barns were selected according to previous tropical theileriosis cases, sampling information's and geographical coordinates were summarized in Table 1. Furthermore, engorged *H. scupense* nymphs were collected from walls of some cattle breeding barns between October and November 2020 from El Hissiene site (upper semi-arid stage, Ariana, Tunisia: N37˚0'14,50469" E10˚10'18,06892).

### Ethical statement

The animals used in this study belonged to farmers who consented to their sampling. The animals were gently restrained by their owner to collect ticks, in the same way as for a routine clinical examination. Invasive sampling and tranquilizers were not used in our work. Furthermore, the sampling was supervised by veterinarians and veterinary technical staff of the National School of Veterinary Medicine, Sidi Thabet, Tunisia. For these reasons, the present study followed the guidelines for the care and use of animals of the National School of Veterinary Medicine, Sidi Thabet, Tunisia, and required no ethical approval.

**Table 1. Bioclimatic stage, governorate, location, delegation, farms visited, surveyed cattle and ticks collected.**

| Bioclimatic stage | Governorate | Location | Delegation | Farms | Cattle | Ticks |
|---|---|---|---|---|---|---|
| Upper semi-arid | Bizerte | N37˚4'5,606" E9˚55'18,90833 | Besbesia | 3 | 68 | 79 |
| Upper semi-arid | Ariana | N37˚0'14,50469" E10˚10'18,06892 | Hessiènne | 4 | 36 | 56 |
| Subhumid | Bizerte | N37˚11'27,79825" E 10˚1'42,14406 | El Alia | 1 | 7 | 20 |
| Subhumid | Bizerte | N 37˚11'25,9242" E 10˚5'1,70138" | Aousseja | 1 | 16 | 33 |
| Upper semi-arid | Seliana | N36˚20'16,65114" E 9˚7'49,99084" | El krib | 5 | 40 | 45 |
| Sub-humid | Bizerte | N 37˚2'16,69924" E 9˚24'29,45606" | Ghezala | 7 | 36 | 151 |
| Humid to sub-humid | Jendouba | N36˚38'54,26635" E 8˚37'49,56319" | Fernena | 7 | 56 | 32 |
| Humid to sub-humid | Beja | N 36˚47'49,0709" E 9˚8'47,70218" | Amdoun | 4 | 25 | 27 |
| Humid to sub-humid | Beja | N36˚50'11,08694" E 9˚5'14,64418" | Beja | 3 | 10 | 7 |
| Humid | Beja | N36˚51'53,61602" E 9˚9'46,29971" | Nefza | 4 | 32 | 34 |
| Upper Arid | Kairouan | N 35˚35'35,488" E 9˚30'9,82742" | El Ala | 8 | 30 | 65 |
| Upper Arid | Kasserine | N35˚15'20,78406" E 9˚4'54,57695" | Sebitla | 7 | 35 | 53 |
| Semi-arid | Zaghouan | N36˚26'10,44344" E 9˚50'1,46022" | Fahs | 9 | 42 | 13 |
| Lower semi-arid | Sousse | N35˚55'51,19154" E 10˚29'7,65161" | Kalaa Kbira | 11 | 32 | 30 |
| Humid | Jendouba | N36˚56'44,03688" E 8˚45'5,30809" | Tabarka | 8 | 25 | 131 |

## Tick species identification

Tick species were first identified morphologically according to the key of Walker et al. [27]. Subsequently, molecular characterization was done using the mitochondrial 16S and 12S rRNA genes for all species and in particular on fully engorged females and nymphs. All sequences were compared with GenBank data using BLAST analysis. We have considered that ticks of the same species are sharing a similarity of ≥97% [28]. All DNA samples were stored at -20°C until their use.

## Genomic DNA extraction and pooling

We have first sterilized the exoskeleton of ticks to eliminate surface contaminants using commercial bleach at 1% for 30 seconds followed by three washes with ultra-pure water according to the protocol of Binetruy et al. [29]. DNA was extracted from individual adult ticks using the DNeasy Blood &Tissue Kit (QIAGEN, Valencia, CA), while DNA extraction from *H. scupense* engorged nymphs and eggs was performed following the Wizard Genomics DNA kit (Promega, Madisson) protocol. The extracted DNA was stored at -20°C for future use.

To study the specific microbial population of ticks, DNA from individual samples was pooled together in equal concentrations for each species and stage. This approach was used in this first investigation to reduce the financial and technical burden of the NGS analysis.

A total of six pools were prepared for next-generation sequencing composed as follow: one pool composed of 60 nymphs of *H. scupense*, one pool of *H. scupense* eggs and 4 pools of adult ticks composed each one of 30 *H. scupense* females, 30 *H. scupense* males, 30 *H. marginatum* adults and 30 *H. excavatum* adults.

## Library preparation for metagenomic sequencing

DNA was quantified using the Qubit dsDNA HS Assay kit (Invitrogen). Library preparation was carried out using NextEra cd index prep with Illumina Miseq. Briefly, 12.5 ng of DNA from each sample were amplified using a v3-v4 primer of 16s rDNA. Samples were dual indexed the pooled and quantified as the final library. An amount of 4 nM of the pooled library was normalized and denatured with NaOH, then samples were loaded on the Illumina Miseq instrument for sequencing.

## Metagenomic analysis

Metagenomic analysis was carried out using qiime2 (Quantitative Insights Into Microbial Ecology) [30] (version 2020.6) from (https://qiime2.org/), raw reads were filtered and adapters were trimmed, reads were denoised using deblur and the table of features was generated, the subsampling of features for the alpha diversity analysis was carried out after the rarefaction of samples by 5624 features. BarPlots of relative abundance were created using the Phyloseq package and Vagans library on Rstudio and the Piecharts were created with Plotly after treatment with Biom algorithm.

## Prevalence and transmission of symbionts

Independent *Francisella* and *Rickettsia* endosymbionts identification was performed using PCR by amplifying the rRNA 16S and glta gene with specific primers (Table 2). PCR conditions set with denaturation step at 94°C for 7 min followed by 40 cycles at 94°C for 45s, 62°C for 30s and 72°C for 45s with a final extension at 72°C for 7 min. The negative DNA template for symbiont was verified systemically by the amplification of ticks' 18S rRNA gene using

**Table 2. Targeted genes, primers, and nucleotide sequences per tick species.**

| Species | Target gene | Primers | Nucleotide sequence (5′–3′) | Reference |
|---|---|---|---|---|
| *Rickettsia* endosymbiont | Citrate synthase (gltA) | gltA-F gltA-R | TCCTACATGCCGACCATGAG AAAGGGTTAGCTCCGGATGAG | [31] |
| *Francisella* endosymbiont | 16S rRNA | Fran16S-F Fran16S-R | CAGGACTAGCTTATAGTTGCTG CATCTGCGACAGCCTAAAAGC | [19] |

universal primers of eucaryotic. PCR products of glta and 16S were purified and sequenced from both directions. Sequences were verified using BLAST.

To compare the prevalence of symbionts between sex and life stage of *H. scupense* ticks, the test chi 2 was performed. Results were considered significant at 5% threshold.

### Maternal transmission assay

A total of 9 fully engorged live females of *H. scupense*, *H. excavatum* and *H. marginatum* were collected from cattle. Engorged ticks were separately incubated at 27˚C and 90% relative humidity until the end of oviposition. DNA was extracted from 300 eggs of each tick and stored at -20˚C until further use.

## Results

### Tick species identification

Due to the sampling design *Hyalomma scupense* was the main collected species, representing more than 40.0% of collected ticks (n = 776), *Haemaphysalis punctata* (18.81%), followed by *Hyalomma marginatum* (17.80%) then *Hyalomma excavatum* (17.36%), *Rhipiciphalus* spp. (6.94%), and finally *Hyalomma impeltatum* (0.14%). Morphologically identified specimens of *H. scupense*, *H. marginatum* and *H. excavatum* used in our study were later confirmed with molecular identification and sequencing using 16S and 12S rRNA genes.

### Microbial diversity in ticks

We have essentially focused on the characterization of the microbial composition and microbiota diversity, richness, and their variation between male and female as well as between different life stages of *H. scupense*. We have also carried out a comparative study with *H. marginatum* and *H. excavatum* which adult stage commonly infest Tunisian cattle in Tunisia. In this regard, 16S rRNA (V3-V4 region) sequencing and metagenomic analysis were established. The relevant accession numbers used to access to our data are available on the following links https://www.ncbi.nlm.nih.gov/sra/PRJNA791601 (for all data).

Denoising reads with deblurring were generated 97081 features where the greatest number of reads was observed in female *H. scupense* and the lowest number of reads was detected in male *H. scupense*. The denoising allowed the selection of 124 representative sequences Operational Taxonomic Unit (OTU) with a mean length of 148 bp. The taxonomic assignment of OTUs was carried out using the Silva 16S database with a cutoff of 99%, the classifier was trained using the qiime feature-classifier algorithm for V3-V4 primers.

Bacteria profiling from different identified livestock species showed that proteobacteria were the dominant phylum with more than 97% of relative percent abundance (RPA) followed by Actinobacteria, Firmicutes, and Deinococcota. Overall, 16 bacterial families were detected among those Rickettsiacea, Francisellacea, Enterobacteriaceae, Bacillaceae, and Brevibacillacea were the most predominant families in the examined tick species (Fig 1). In the male of *H. scupense*, Enterobacteriaceae was the most prevalent bacterial family with 69.5% RPA followed by

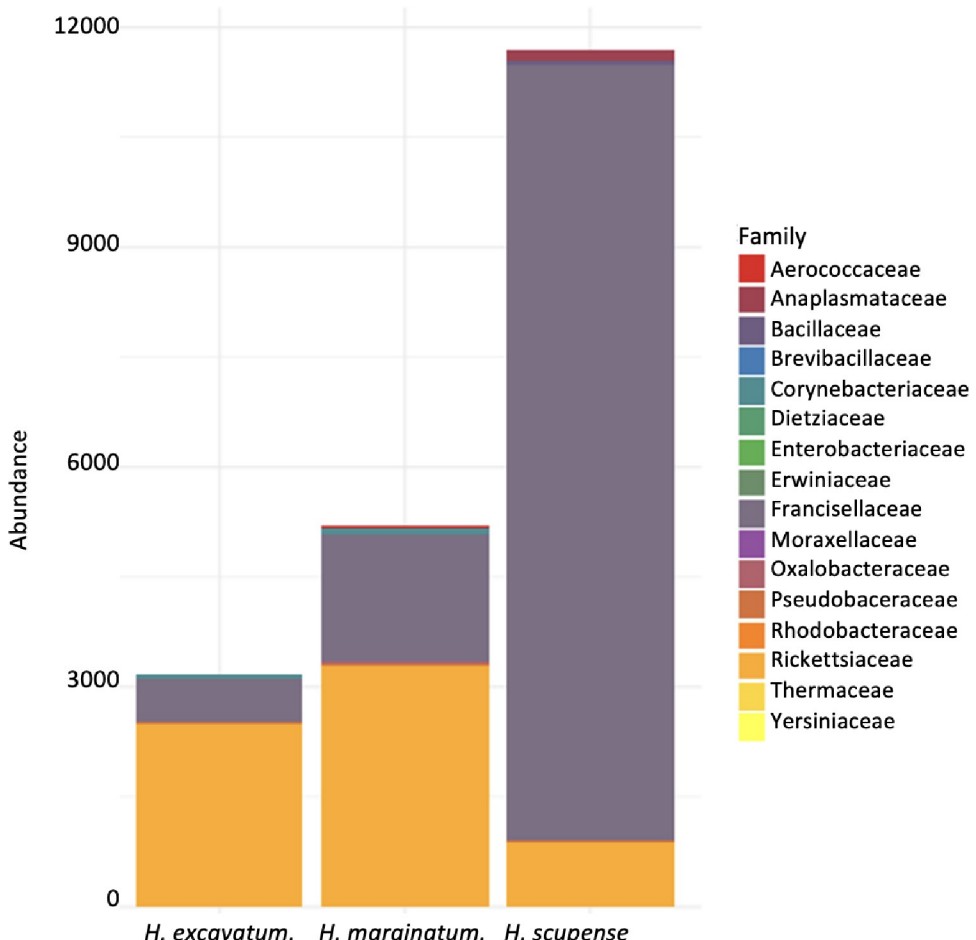

**Fig 1. Family level taxonomic composition of *Hyalomma excavatum*, *H. marginatum* and *H. scupense* adult ticks.**

Rickettsiaceae 22.9% and Francisellacea 2.56%. However, Francisellacea was the dominant family in the female of *H. scupense* with 78.8% of RPA followed by Rickettsiaceae with 19.9% of RPA. In the two other *Hyalomma* species, Rickettsiaceae were the most abundant followed by Francisellacea.

Three symbiotic bacterial genera were detected in the analyzed ticks, these were *Francisella*, *Rickettsia and M. mitochondrii*. *Francisella* and *Rickettsia* genera were detected in all three species of adult *Hyalomma*, they were also present in all the stages of *H. scupense* (Fig 2), however at different RPA. The highest RPA of *Francisella* was recorded in the females of *H. scupense* with an RPA of 78.8%, followed by *H. marginatum* sample with an RPA of 37.9%. However, the lowest rate was observed in the male of *H. scupense*. The Rickettsia genera make up the vast majority of *H. excavatum* with 70.8% RPA, it was also detected at a high abundance of 34.5% in *H. marginatum*. *M. mitochondrii* was highly present *in H. marginatum and H. excavatum* with an RPA of 34.5 and 17.3% respectively.

We have also detected some genera of pathogen bacteria for instance; *Ehrlichia, Erwinia, Dietzia, Escherichia, Shigella, Pseudomonas*, as well as the two other symbiotic/pathogenic bacteria: *Francisella* and *Rickettsia*. To examine the possible difference in bacterial population structure between species, sexes, and life stages, alpha diversity parameters were calculated (Table 3 and Fig 3).

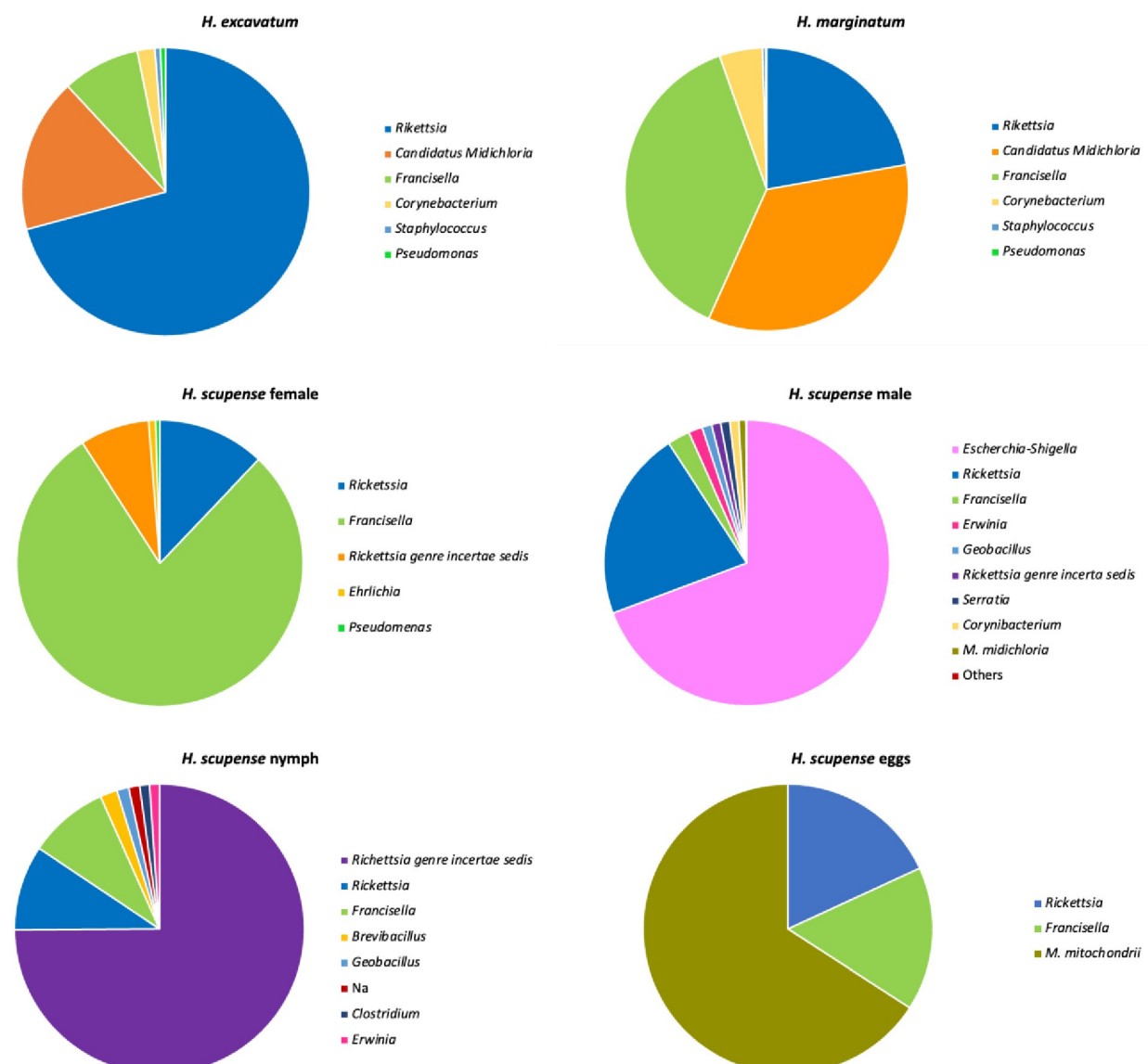

**Fig 2. Relative abundance (%) of bacterial genera according to tick species and life stage of *Hyalomma scupense*.** Organisms listed in the legend appear from top to bottom in order of highest relative abundance.

Notably, there was a significant difference between adult species, *H. excavatum* and *H. marginatum* samples have a significantly more diverse microbiome than *H. scupense* as indicated by the Chao and Simpson indexes (Table 3 and Fig 3). *H. scupense*, female ticks have less microbial diversity than males, furthermore, there was a significant difference between the relative abundance of the bacterial genus and species detected in female versus male ticks (p<0.05), indeed, 40 species were detected in males while 35 species were recorded in females (Fig 4). Indeed, *Francisella* endosymbionts constituted by far a dominant percentage of the microbiome of female ticks comparatively to male ticks, 78.8% and 2.51% respectively. Although a significant difference between different stages was detected since adult ticks exhibit a greater extent of microbial diversity. We then examined the microbial richness using the Shannon diversity index (Table 3) and similar trends were observed with the Chao index

**Table 3. Alpha diversity indexes.**

| Sample | Obs | Chao1 | se.chao1 | ACE | se.ACE | Shannon | Simp | InvSimp | Fisher |
|---|---|---|---|---|---|---|---|---|---|
| *H. scupese* female | 12 | 12,00 | 0,24 | 12,46 | 1,73 | 0,96 | 0,47 | 1,87 | 1,61 |
| *H. scupense* male | 15 | 15,00 | 0,00 | 15,00 | 1,55 | 1,12 | 0,57 | 2,35 | 2,08 |
| *H. scupense* nymph | 23 | 23,25 | 0,73 | 23,67 | 2,29 | 1,23 | 0,55 | 2,25 | 3,43 |
| *H. excavatum* | 15 | 15,00 | 0,00 | 15,00 | 1,93 | 1,29 | 0,67 | 3,00 | 2,08 |
| *H. marginatum* | 15 | 15,00 | 0,00 | 15,00 | 1,93 | 1,65 | 0,76 | 4,16 | 2,08 |
| *H. scupense* eggs | 10 | 10,50 | 1,28 | 11,79 | 1,37 | 1,40 | 0,70 | 3,37 | 1,30 |

because *H. scupense* samples have less rich diversity than the other two species. However, in the case of *H. scupense*, nymphs were by far the richest in diversity relative to the two other life stages.

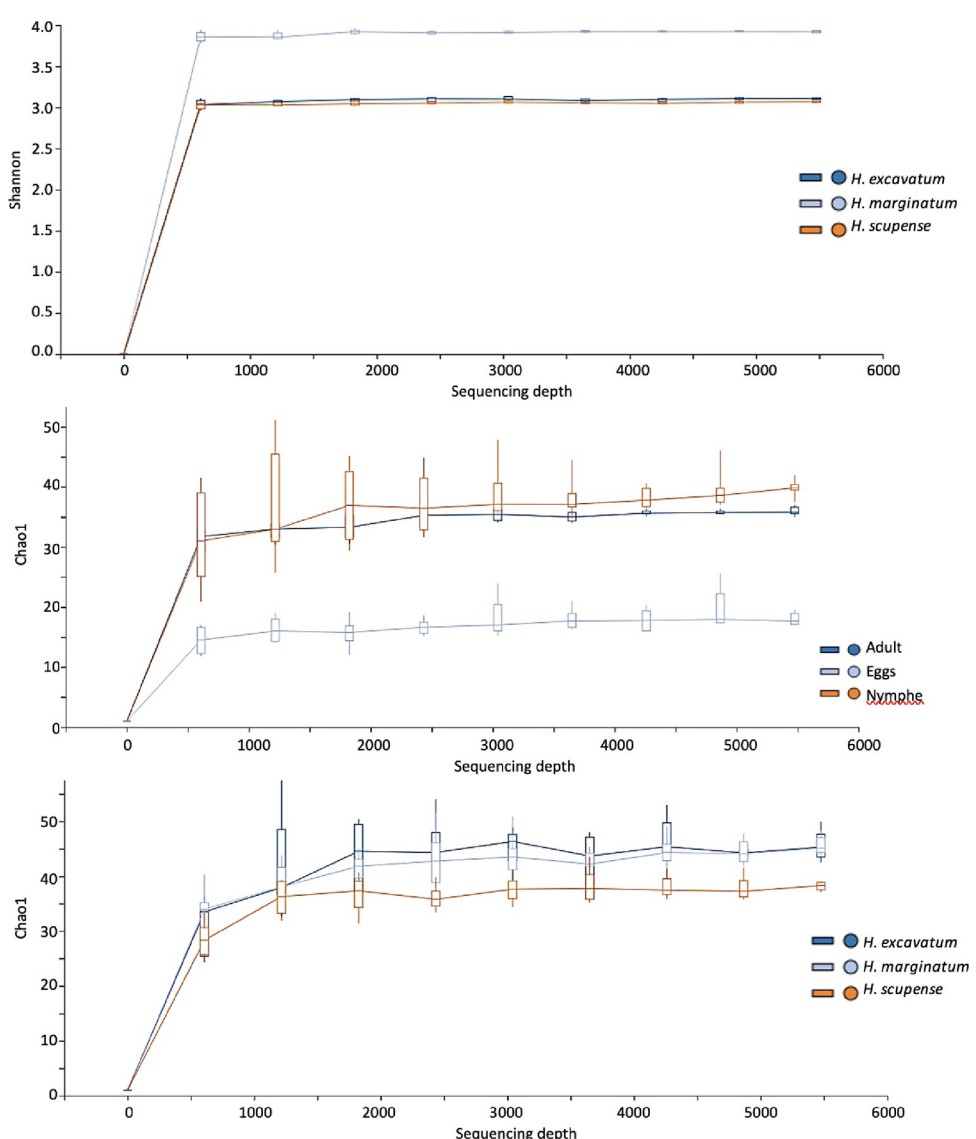

**Fig 3. Alpha diversity and richness parameters of *Hyalomma scupense* microbiota.** a: Shannon alpha diversity between species; c: Chao alpha diversity between species; b: Chao alpha diversity between life stages.

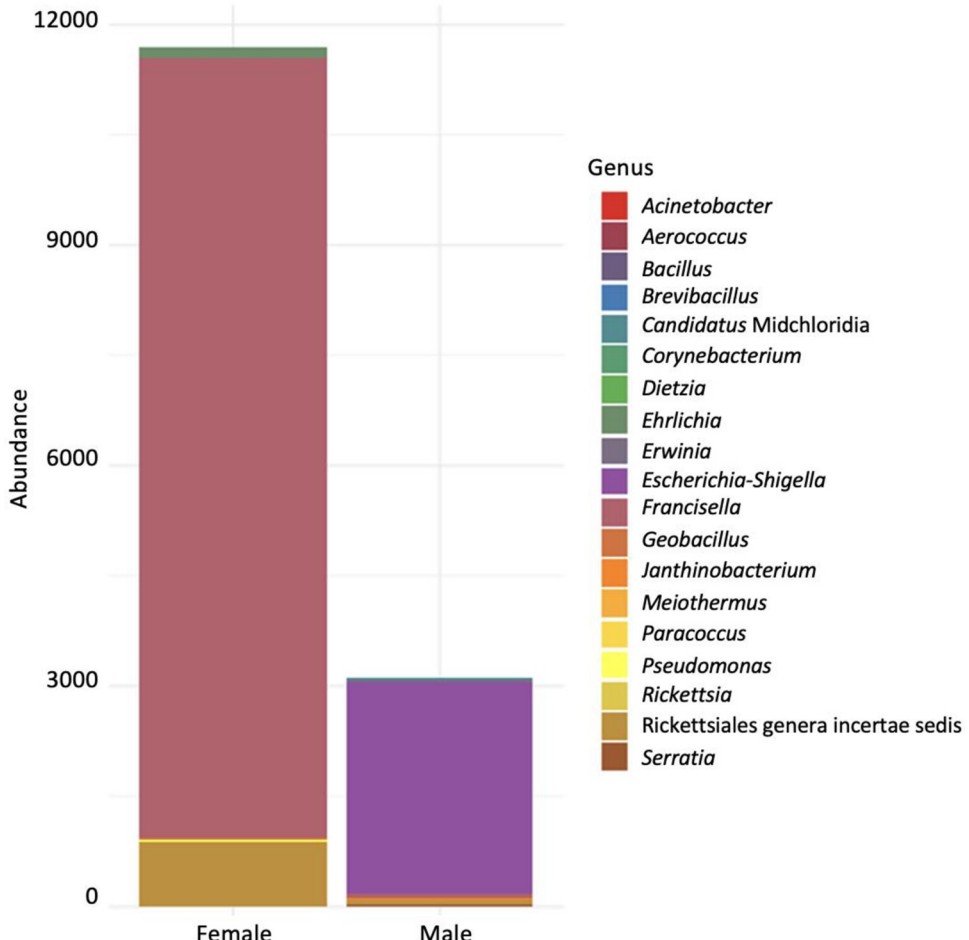

**Fig 4. Genus's level taxonomic composition of male and female *Hyalomma scupense* ticks.**

## Maternal transmission

To test whether *Francisella* and *Rickettsia* endosymbionts have a maternal transmission. Samples of eggs of nine fully engorged females of *H. scupense*, *H. excavatum* and *H. marginatum* ticks were screened by PCR amplification. PCR product was sequenced and revealed that all the pools of eggs of each female were positive for the two symbionts with an overall transovarial transmission efficiency of 100%. Although using NGS, we have detected the presence of these two symbionts in *H. scupense* eggs at an RPA of 16% and 18.1% for *Francisella* and *Rickettsia* respectively. However, the most dominant endosymbiont detected in *H. scupense* eggs was *M. mitochondrii* with 65.5% RPA (Fig 2). This endosymbiont was not detected in *H. scupense* adult ticks, while as reported it was highly represented in adults of *H. marginatum* and to a lesser extent in those of *H. excavatum*.

## Prevalence of symbionts

To assess the prevalence of symbionts in *H. scupense* nymphs and adult tick stages, as well as the effect of gender, we first sequenced the PCR product for each symbiont to validate the sequences of the symbionts. Sixteen engorged nymphs were tested and the results show that 45% of them carry *Francisella* endosymbiont while 22.22% of them harbor *Ricketessia*

endosymbiont. For adult tick, PCR amplification of 40 males *H. scupense* revealed that 35.13% harbor *Francisella* endosymbiont. Interestingly, 80% of half-engorged females were positive for *Francisella* endosymbiont while 66,66% of them were positive with *Rickettsia* endosymbiont. Interestingly for *Francisella* endosymbiont, we have detected a significant difference between adult and nymphal stages as well as between males and females ($p<0.05$).

## Discussion

Previous surveys on ticks microbiota have shown carriage of multiple symbionts which are very important for their physiology, fecundity, and vector competence [32–34]. Currently, several studies were carried out on microbial populations in ticks such as *Ixodes*, *Rhipiciphalus*, *Haemaphysalis*, *Amblyomma*, *Dermacentor* [35–41]. However, little is known about the diverse microbiome of the tick genus *Hyalomma* associated with cattle. Moreover, there is no information regarding the microbial population structure of the most dominant Tunisian cattle tick *H. scupense* despite its veterinary and economic importance. It is of great interest to know crucial details on the microbial composition and their symbionts to improve our knowledge about tick bacterial communities. Therefore, this investigation may shed light on the vectorial competency of these vectors to transmit pathogens to vertebrate hosts. We present here the first characterization of the whole microbial communities of the monotropic cattle tick *H. scupense* according to the stages and sex and its comparison with the microbial diversity of adults *H. scupense* tick of two other dominant *Hyalomma* species which adult stages are infesting cattle, namely *H. excavatum* and *H. marginatum*.

Next generation sequencing of the hypervariable V3-V4 region of bacterial 16S rRNA gene was performed using *Hyalomma* specimens with Illumina Miseq. Similar to other studies, we found that the most dominant phylum was Proteobacteria. Additionally, the bacterial microbiota of *Hyalomma* was dominated by three principal genera; namely the two symbiotic/pathogenic bacteria *Francisella* and *Rickettsia*, and the endosymbiotic *M. mitochondrii*. Previous studies on other genera like *Rhipicephalus* have shown that they were mainly infected with *Coxiella* and *Rickettsia* with an association of *Coxiella*-like endosymbionts with females [37, 42]. A more recent study on the microbiome of *Hyalomma* species infesting livestock in Pakistan showed that *Francisella*, *Rickettsia*, and *Coxiella* were the most dominant tick bacterial identified species [43, 44]. Nevertheless, our results correlate with another survey on ticks in Turkey, where *Coxiella* was not detected in *Hyalomma* samples, while *Francisella* and *Rickettsia* predominate in the microbiome of *H. excavatum* and *H. marginatum* [28]. In our study, it was not entirely surprising that we detected the genus *Francisella* at high relative abundance associated with the absence of *Coxiella* endosymbiont, despite the higher rate of transovarial transmission and the abundance of *Coxiella* endosymbiont in several other tick genera. This endosymbiont was detected using NGS in a large variety of hard ticks, including *Amblyomma americanum* [45], *Haemaphysalis longicornis* [35], *Ornithodoros amblus* [46], *Rhipicephalus microplus* [47], and *Rhipicephalus sanguineus* [37]. Our findings are similar to those of Duron et al. (2015) who detected at low frequency *Coxiella* endosymbiont in approximately one-third of their targeted species [47]. Furthermore, Gehrart and his team suggested that *Francisella* has replaced Coxiella in the Gulf Coast tick, *Amblyomma maculatum* [48].

We suggest that the lack of detection of *Coxiella* infection in our samples and the abundance of another maternally inherited endosymbiont may be due to the replacement and competition of this ancient endosymbiont with other maternally inherited endosymbionts. In our case, we have revealed the presence of three other genera of maternally inherited bacteria: *Francisella*, *M. mitochondrii* and *Rickettsia* endosymbiont in all tick specimens. The high distribution of *Francisella* among all our tick species suggests that they may have independently

replaced *Coxiella* endosymbiont as an alternative obligate symbiont. The importance of *Francisella* in Tunisian *Hyalomma* ticks is probably the result of the interaction and coevolution between the microbiome community and the tick's ecological modification and restricted diet. Although it is important to emphasize that physiological and nutritional investigation on the symbiotic functions of *Francisella* are required to validate their role as alternative obligate symbionts. Moreover, Genome sequencing of these bacteria demonstrates that they evolved by developing adaptive mechanisms implicated in tick survival however, they have highly conserved the major genes responsible for the synthesis of vitamins B, cofactors, and some amino acids like Coxiella endosymbiont despite their restraint genome [19–21, 24, 48].

Interestingly, we report here, using high throughput sequencing, the first detection of the obligate intracellular mitochondrial tick endosymbiont *M. mitochondrii* in eggs of *Hyalomma* tick with a high abundance of more than 65%, however this density was considerably decreased in the other stages of *H. scupense* (nymphs and adults), along with the increase of other two endosymbionts. The distribution and the transovarial transmission of this intracellular endosymbiont in *H. scupense* suggested that this association might be obligate, and have a potential role in the physiological fitness of *H. scupense* tick. Furthermore, this bacterium is significantly abundant in *H. marginatum* and to a lesser extent in *H. excavatum* with RPA of 34.5 and 17.8% respectively. Recently, Selmi et al. reported that *M. mitochondrii was* prevalent at a rate of 8% in partially fed *H. dromedarii* and *H. impeltatum* ticks collected from dromedaries in Tunisia [49]. A recent study on the detection of pathogens and *M. mitochodrii* in Egyptian cattle ticks using the Reverse Line Blot hybridization (RLB) showed that 11.6% of pools of *H. excavatum* are infected with *M. mitochondrii* while only 2.9% of *R. annulatus* pools are infected by this bacteria [50]. Moreover, the detection of this endosymbiotic bacterium in other hard ticks is very variable [26, 51]. *M. mitochondrii* is well-described in *I. ricinus* [25] where it was detected in the mitochondria of ovaries. These bacteria which are included in the α-Proteobacteria entered the mitochondria of the oocyst cells. About 94–100% of adult females of *I. ricinus* are infected with this bacterium with a 100% transovarial transmission rate [25].

Additionally, we have studied various aspects of microbial diversity including species, sex, and life stage. The investigation of the microbial diversity with *Haylomma* species indicates that the adults of *H. marginatum* and *H. excavatum* exhibits more diverse and richer microbiota than *H. scupense*. Previous microbial diversity studies in three *Hyalomma* species collected in Turkey also indicated that *H. excavatum* and *H. marginatum* harbor more diverse microbiota than *H. aegyptium* [28]. This diversity is influenced by several determinants, mainly, environmental factors like habitat [52], geographical dispersion [53], bioclimatic conditions [54], sample location [55], soil and plant associated bacteria [56], and tick's host [38, 57].

*H. marginatum* and *H. excavatum* are probably exposed to more complex environmental conditions as they pass through different hosts during their development particularly at juvenile stages [58]. Moreover, it is well known that blood meals have a strong impact on tick microbial diversity, composition, and species richness as mentioned by Swei and Kwan [38]. Indeed, ticks acquire more microorganisms, including pathogens, from hosts during their blood meals. More diverse hosts and larger numbers of vertebrate hosts hypothetically drive to more diverse microbiota [59]. This is probably the case for the outdoor *H. marginatum* and *H. excavatum* ticks, indeed, these tick juveniles feed, according to their tropism, on a variety of small terrestrial mammals and birds, whilst their adults can engorge on a variety of large herbivores such as goats, sheep, cattle, horses, camels [60, 61]. In contrast, *H. scupense*, exhibits a domestic behavior, and cattle are by far the almost exclusive host for adults and juveniles in Tunisia [5, 62]. Other studies found that the use of different tick species feeding on the same host don't even have the same microbiomes, suggesting, therefore, that the tick microbiome is,

to a certain degree, governed by species specific determinants [63]. Microbial composition was also affected by sex. Females *H. scupense* microbiota was less diverse and rich as compared to males, but inherited a high relative abundance of endosymbionts than males especially their OTUs number. We also found a higher prevalence of *Francisella* in females, especially in engorged females (100%). Similar results were reported in other studies that typically [42, 64, 65] In the same way, Van Treuren and his team found that females show less diverse microbiota than males [53]. The increased RPA of *Rickettsia* and *Francisella* endosymbiont that we recorded in females relative to males could be an adaptation to reproductive requirements and transovarial transmission. In addition, we demonstrated efficient maternal transmission of these two endosymbionts.

According to our metagenomic analysis, we found a significant loss of bacterial species richness in adults when compared to nymphs. *Francisella* and *Rickettsia* become dominant at older life stages correlated with the loss of microbiome richness. Our results are similar to another study on the microbiome of *Ixodes pacificus* [38]. This change is obviously due to the variation on tick activity and their metabolic function between their different life stages and sex. Indeed, tick's obligatory endosymbiont have a crucial role in its development, nutrition, reproductive fitness, and tick adaptation to new environmental conditions in the tick life cycle [11, 59]. Furthermore, it is also important to consider the potential role of the environmental conditions and the difference of the ecological niches between species during their life cycle in the acquisition and the expression of microbiota like for instance the case of the domestic *H. scupense* with the outdoor ticks *H. marginatum* and *H. excavatum*.

Our results on symbionts of *Hyalomma* ticks may contribute to improve our knowledge on the interaction between tick bacterial communities and their hosts opening the way to develop subsequently applied research targeting the development of new control options against ticks and tick-borne pathogens. Bacterial endosymbionts may affect tick's physiology and their reproductive fitness, they may also influence tick's vectorial capacity for transmitted pathogens, and finally, they may interact with the tick hosts with potential veterinary and zoonotic outcomes in particular for *Francisella* and *Rickettsia* bacteria.

## Conclusion

Overall, the analysis of the bacterial diversity within three *Hyalomma* species reveals that *H. marginatum* and *H. excavatum* have greater diversity than *H. scupense.* Furthermore, microbial diversity and composition vary according to the tick's life stage and sex in the specific case of *H. scupense*. The endosymbionts *Francisella*, *M. mitochondrii* and *Rickettsia* were shown to be the most prevalent in *Hyalomma*. According to our findings, all studied *Hyalomma* tick species possess zoonotic bacteria genera and could potentially operate as vectors for various zoonoses. *Rickettsia* and *Francisella* are the most common zoonotic bacteria found in *Hyalomma* ticks.

This study provides general information about microbial communities in *Hyalomma* ticks and their endosymbionts. This information provides critical clues for future studies aimed at the prevention and control of neglected tick and tick-borne diseases in the area using innovative approaches.

## Acknowledgments

The authors thank Mr. Limam Sassi and Mr. Taoufik Lahmar for their technical support. The authors acknowledge also all the veterinarians for their valuable help and all the farmers that participated to this study.

## Author Contributions

**Conceptualization:** Soufiene Chaari, Mourad Rekik, Hadda-Imene Ouzari, Tarek Hajji, Mohamed Aziz Darghouth.

**Formal analysis:** Hayet Benyedem, Abdelmalek Lekired, Moez Mhadhbi, Mokhtar Dhibi, Rihab Romdhane, Tarek Hajji.

**Funding acquisition:** Soufiene Chaari, Mourad Rekik.

**Investigation:** Hayet Benyedem, Mokhtar Dhibi, Rihab Romdhane, Tarek Hajji.

**Methodology:** Moez Mhadhbi.

**Resources:** Mokhtar Dhibi, Rihab Romdhane.

**Software:** Abdelmalek Lekired.

**Supervision:** Tarek Hajji, Mohamed Aziz Darghouth.

**Validation:** Moez Mhadhbi, Soufiene Chaari, Mourad Rekik, Hadda-Imene Ouzari, Tarek Hajji, Mohamed Aziz Darghouth.

**Visualization:** Soufiene Chaari, Mourad Rekik, Hadda-Imene Ouzari, Mohamed Aziz Darghouth.

**Writing – original draft:** Hayet Benyedem.

**Writing – review & editing:** Abdelmalek Lekired, Moez Mhadhbi, Mourad Rekik, Hadda-Imene Ouzari, Tarek Hajji, Mohamed Aziz Darghouth.

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
