## [Decision Letter · Decision Letter 0]

26 Oct 2021

PONE-D-21-29206First insights into the microbiome of Tunisian Hyalomma ticks gained through next-generation sequencing with a special focus on H. scupensePLOS ONE

Dear Dr. Mohamed Aziz Darghouth,

Thank you for submitting your manuscript to PLOS ONE. After careful consideration, we feel that it has merit but does not fully meet PLOS ONE’s publication criteria as it currently stands. Therefore, we invite you to submit a revised version of the manuscript that addresses the points raised during the review process.

ACADEMIC EDITOR: It is recommended to do a statistical analysis of the manuscript results==============================

We look forward to receiving your revised manuscript.

Kind regards,

Shawky M. Aboelhadid, PhD

Academic Editor

PLOS ONE

Journal Requirements:

2. In your Methods section, please provide additional location information, including geographic coordinates of your field collection site if available.

3. We note that you are reporting an analysis of a microarray, next-generation sequencing, or deep sequencing data set. PLOS requires that authors comply with field-specific standards for preparation, recording, and deposition of data in repositories appropriate to their field. Please upload these data to a stable, public repository (such as ArrayExpress, Gene Expression Omnibus (GEO), DNA Data Bank of Japan (DDBJ), NCBI GenBank, NCBI Sequence Read Archive, or EMBL Nucleotide Sequence Database (ENA)). In your revised cover letter, please provide the relevant accession numbers that may be used to access these data. For a full list of recommended repositories, see http://journals.plos.org/plosone/s/data-availability#loc-omics or http://journals.plos.org/plosone/s/data-availability#loc-sequencing.

 “HB

CGIAR Research Program on Livestock (CRP Livestock)

Laboratory of epidemiology of enzootic infections in herbivores in Tunisia: application to control (LR16AGR01), Ministry of Higher Education and Scientific Research, Tunisia

PromEssE - ANPR MOBIDOC, Ministry of Higher Education and Scientific Research, Tunisia”

“This study was financially supported by the ICARDA - CGIAR Research Program on Livestock (CRP Livestock), and the "Laboratory of epidemiology of enzootic infections in herbivores in Tunisia: application to control" (LR16AGR01) funded by the Ministry of Higher Education and Scientific Research, Tunisia. This study is also carried out under the MOBIDOC scheme, funded by The Ministry of Higher Education and Scientific Research through the PromEssE project and managed by the ANPR”

“HB

CGIAR Research Program on Livestock (CRP Livestock)

Laboratory of epidemiology of enzootic infections in herbivores in Tunisia: application to control (LR16AGR01), Ministry of Higher Education and Scientific Research, Tunisia

PromEssE - ANPR MOBIDOC, Ministry of Higher Education and Scientific Research, Tunisia”

7. In your Data Availability statement, you have not specified where the minimal data set underlying the results described in your manuscript can be found. PLOS defines a study's minimal data set as the underlying data used to reach the conclusions drawn in the manuscript and any additional data required to replicate the reported study findings in their entirety. All PLOS journals require that the minimal data set be made fully available. For more information about our data policy, please see http://journals.plos.org/plosone/s/data-availability.

8. We note that you have stated that you will provide repository information for your data at acceptance. Should your manuscript be accepted for publication, we will hold it until you provide the relevant accession numbers or DOIs necessary to access your data. If you wish to make changes to your Data Availability statement, please describe these changes in your cover letter and we will update your Data Availability statement to reflect the information you provide.

9. PLOS requires an ORCID iD for the corresponding author in Editorial Manager on papers submitted after December 6th, 2016. Please ensure that you have an ORCID iD and that it is validated in Editorial Manager. To do this, go to ‘Update my Information’ (in the upper left-hand corner of the main menu), and click on the Fetch/Validate link next to the ORCID field. This will take you to the ORCID site and allow you to create a new iD or authenticate a pre-existing iD in Editorial Manager. Please see the following video for instructions on linking an ORCID iD to your Editorial Manager account: https://www.youtube.com/watch?v=_xcclfuvtxQ

10. We note that Figure 1 in your submission contain map images which may be copyrighted. All PLOS content is published under the Creative Commons Attribution License (CC BY 4.0), which means that the manuscript, images, and Supporting Information files will be freely available online, and any third party is permitted to access, download, copy, distribute, and use these materials in any way, even commercially, with proper attribution. For these reasons, we cannot publish previously copyrighted maps or satellite images created using proprietary data, such as Google software (Google Maps, Street View, and Earth). For more information, see our copyright guidelines: http://journals.plos.org/plosone/s/licenses-and-copyright.

Reviewers' comments:

Reviewer's Responses to Questions

**Comments to the Author**

1. Is the manuscript technically sound, and do the data support the conclusions?

Reviewer #1: Yes

Reviewer #2: Yes

2. Has the statistical analysis been performed appropriately and rigorously? 

Reviewer #1: Yes

Reviewer #2: No

3. Have the authors made all data underlying the findings in their manuscript fully available?

Reviewer #1: Yes

Reviewer #2: No

4. Is the manuscript presented in an intelligible fashion and written in standard English?

Reviewer #1: Yes

Reviewer #2: Yes

5. Review Comments to the Author

Reviewer #1: For the authors:

Thank you for this research, which deals with very interesting subject with little data on ticks and its symbiotic and commensal/pathogenic bacteria.

I have some comments/remarks to the respected authors. As follow:

I- Generally:

• Please add abbreviation list for all abbreviations you used e.g. rRNA, 16S, 25S, V3-V4, OUT, RPA, NGS.

• 'Candidatus Midichloria mitochondrii' please review this bacterium species scientific name and then after unify witting name and style of this bacterium throughout the manuscript.

II- Other comments: many typo errors, please revise the manuscript well

Abstract:

• P7: next generation

The analysis…

….for 16S RNA of?????

Hyalomma spp. or use H. scupense

Common bacterial genera founf in Hyalomma ticks

Introduction

• P8 : Hyalomma spp. is the most prevalent ticks infesting cattle

Materials and methods

• P10: 'adult' not italic

Cattle breeding barns

16S and 25S rRNA genes (repeat in P13)

'bleach' please specify the material you used

• P11: 'tick's egg' replace by Hyalomma spp. egg

• P11 and P13: in table 2 mentioned that you used a specific primer to amplify Cytochrome b mtDNA gene of Theileria annulata from the collected ticks. Interestingly, I did not any results or discussion on the usage of this primer, why??!!, although in the introduction section and study design, you emphasized cattle theileriosis and ticks role in it also, mentioned that ticks were collected from sites with pervious history of theileriosis.

• P12: 9 instead of nine

Results

• P13: spp. not italic

• P14: However, Francisellacea was the dominant family in the female of H. scupense with 78.8% of…..

'adult' not italic

Fig.2: adult ticks

• P15: Fig 4. Hyalomma scupense

• P16: ' suggesting the potential reproductive role of those symbionts and their correlation with the tick stage. ' please omit or rephrase or remove to discussion section as no suggestions in the results. Results are based on facts not suggestions.

Discussion

• P18: Selmi et al. reported

• P19: 'RA'??!! you mean RPA or another meaning?

Dear respected authors, thank you again.

Regards.

Reviewer #2: Dear Authros, The article is written well but there are some shortcomings that can be incorporated to improve it

1. There are formatting errors (pdf file is attached)

2. There need statistical analysis like regression to find prevalence and correlation with relation to area, tick stage and gender of the tick

3. There is a recent published article related to your article which should be incorporated in discussion

Al-Hosary A, Răileanu C, Tauchmann O, Fischer S, Nijhof AM, Silaghi C. Tick species identification and molecular detection of tick-borne pathogens in blood and ticks collected from cattle in Egypt. Ticks Tick Borne Dis. 2021 May;12(3):101676. doi: 10.1016/j.ttbdis.2021.101676. Epub 2021 Jan 26. PMID: 33540276.

4. There is need to change the figure 3 and it is suggested to keep the colours referring to each organism must be same in the whole figure.

Regards

6. PLOS authors have the option to publish the peer review history of their article (what does this mean?). If published, this will include your full peer review and any attached files.

Reviewer #1: **Yes: **Khaled Sultan

Reviewer #2: **Yes: **Prof Muhammad Imran Rashid

---

## [Author Response · Author response to Decision Letter 0]

16 Feb 2022

Reviewer #1: 

I- General comments:

• Please add abbreviation list for all abbreviations you used e.g. rRNA, 16S, 25S, V3-V4, OUT, RPA, NGS.

The following abbreviation list was added, please refer to lines 361-369

Abbreviation list:

rRNA: Ribosomal ribonucleic acid 

16S: Gene coding for the small ribosomal 16 RNA gene

12S: gene coding for the small ribosomal 12 RNA gene 

V3-V4: the hypervariable regions of the 16S rRNA gene

OUT: Operational Taxonomic Unit

RPA: Relative Percent Abundance 

NGS: Next Generation Sequencing 

M. mitochondrii: Midichloria mitochondrii

• 'Candidatus Midichloria mitochondrii' please review this bacterium species scientific name and then after unify witting name and style of this bacterium throughout the manuscript.

Verified and standardized as Midichloria mitochondrii from NCBI Taxonomy Browser, all over the MS

II- Other comments: many typo errors, please revise the manuscript well

The manuscript was revised and typo errors were corrected

III- Specific comments

Abstract:

• Point 7 (P7): next generation

Corrected as required, please refer to page 2 line 3

The analysis…

corrected (T in capital), Page 2 Line 4

….for 16S RNA of?????

Corrected as follows: for the 16S rRNA (V3-V4 region) of tick bacterial microbiota and…, please refer to page 2 lines 3 and 4

Hyalomma spp. or use H. scupense

corrected as follows: Hyalomma spp, please refer to Page 2 line 8

Common bacterial genera found in Hyalomma ticks

corrected as follows: Accordingly, Hyalomma ticks could represent potential vectors for these zoonotic bacterial agents…, please refer to Page 2 lines 9-10

Introduction

• P8 : Hyalomma spp. are the most prevalent ticks infesting cattle

corrected as required, please refer to page 2 line 21

Materials and methods

• P10: 'adult' not italic

Corrected as required, all over the MS

Cattle breeding barns

Corrected, please refer to Page 3 line 16

16S and 25S rRNA genes (repeat in P13)

Corrected as follows: 16S and 12S rRNA genes, please refer to page 3 line 23 and all over the MS

'bleach' please specify the material you used

The used material was specified as required: commercial bleach, please refer to P4 line 3

• P11: 'tick's egg' replace by Hyalomma spp. Egg

The sentence was changed as follows: one pool of H. scupense eggs and…, please refer to p4 line 12

• P11 and P13: in table 2 mentioned that you used a specific primer to amplify Cytochrome b mtDNA gene of Theileria annulata from the collected ticks. Interestingly, I did not any results or discussion on the usage of this primer, why??!!, although in the introduction section and study design, you emphasized cattle theileriosis and ticks role in it also, mentioned that ticks were collected from sites with pervious history of theileriosis.

The authors agree with reviewer comment, the primers were removed from the table, it was a careless mistake, please refer to table 2

• P12: 9 instead of nine

Done, please refer to p6 line 15. 

Results

• P13: spp. not italic 

Corrected, please refer to p7 line 8 

• P14: However, Francisellacea was the dominant family in the female of H. scupense with 78.8% of…..

Corrected, please refer to p8 line 20

'adult' not italic

Fig.2: adult ticks

Corrected 

Please refer to p8 line 24

• P15: Fig 4. Hyalomma scupense.

corrected in italic, p9 line 18

• P16: ' suggesting the potential reproductive role of those symbionts and their correlation with the tick stage. ' please omit or rephrase or remove to discussion section as no suggestions in the results. Results are based on facts not suggestions.

The text was removed

Discussion

• P18: Selmi et al. reported

Corrected as required, please refer to p12 line 29

• P19: 'RA'??!! you mean RPA or another meaning?

Corrected: RPA, please refer to p14 line 3

Reviewer #2: Dear Authros, the article is written well but there are some shortcomings that can be incorporated to improve it

1. There are formatting errors (pdf file is attached)

All suggestions made by Reviewer #2 were considered. The changes in the MS were put in blue color. 

For the Reviewer #2 questions hand written in the pdf version: 

Q1: Rephrase, no sense (p4, Study area and tick sampling)

The text was modified as follows:

For this purpose, cattle barns were selected according to previous tropical theileriosis cases

Please refer to lines 14 and 15

Q2 Why not 18S rRNA, ticks are eukaryote?

We agree the Reviewer comment, ticks are eukaryotes of course, however, we targeted the mitochondrial 12S and 16S ribosomal RNA (rRNA) genes.

In fact, rRNA 16S and 12S rRNA are the most commonly used markers to study tick’s phylogeny and they have been demonstrated to be able to differentiate the species of some ticks (Anderson et al., 2004; Vial et al., 2006; Chen et al., 2012). Furthermore, the number of annotated sequences of rRNA 16S genes in GenBank was increasing recently providing a strong database for tick’s species.

The text was modified as follows:

…was done using the mitochondrial 16S and 12S rRNA genes…

Please refer to p4 line 23

Q2 : How 6 groups?

Four groups for adults, one for nymphs and one for eggs

In the MS, the sentence war modified as follows:

A total of six pools were prepared for next-generation sequencing composed as follows: one pool composed by 60 nymphs of H. scupense, one pool of H. scupense eggs and 4 pools of adult ticks composed each one of 30 H. scupense females, 30 H. scupense males, 30 H. marginatum adults and 30 H. excavatum adults. Please refer to p5 lines 11-13

Comment 2. There need statistical analysis like regression to find prevalence and correlation with relation to area, tick stage and gender of the tick

We added statistical analysis using chi 2 test to check the significance of the correlation between of symbiont prevalence and sex, and life stage of H. scupense. 

We have studied the prevalence of symbiont only for H. scupense samples. In this study we targeted the symbionts of H. scupence ticks. Moreover, the parameter “Area” was not taken into consideration in the current study

The following text was added

To compare the prevalence of symbionts between sexe and life stage of H. scupense ticks, the test chi 2 was performed. Results were considered significant at 5% threshold.

Please refer to p6 lines 11 and 12

The “Result section” was also modified, and the test result was added. Please refer to p10 line 3 and p11 Line 4.

3. There is a recent published article related to your article which should be incorporated in discussion

Al-Hosary A, Răileanu C, Tauchmann O, Fischer S, Nijhof AM, Silaghi C. Tick species identification and molecular detection of tick-borne pathogens in blood and ticks collected from cattle in Egypt. Ticks Tick Borne Dis. 2021 May;12(3):101676. doi: 10.1016/j.ttbdis.2021.101676. Epub 2021 Jan 26. PMID: 33540276.

We included this article to the discussion section. The following text was added:

Recent study on the detection of pathogens and M. mitochodrii in Egyptian cattle ticks using the Reverse Line Blot hybridization (RLB) showed that 11.6% of pools of H. excavatum are infected with M. mitochondrii while only 2.9% of R. annulatus pools were infected by this bacteria [50]. Please refer to p12 lines 1-4

The citation was added to the reference list at the end of the MS

4. There is need to change the figure 3 and it is suggested to keep the colours referring to each organism must be same in the whole figure.

The fig 3 was changed as required

Rephrase, omit the redundancy, 

discuss more about the probable understanding factors that shape the microbial using more references

The discussion was partially rephrased and the following text was added:

This diversity is influenced by several determinants, mainly environmental factors like, habitat [52], geographical dispersion [53], bioclimatic conditions [54], sample location [55], soil and plant associated bacteria [56], and also the tick hosts [57,58]. 

H. marginatum and H. excavatum are probably exposed to more complex environmental conditions as they pass through different hosts during their development particularly at juvenile stages [59]. Moreover, it is well known that blood meals have a strong impact on tick microbial diversity, composition, and species richness as mentioned by Swei and Kwan [38]. Indeed, ticks acquire more microorganisms, including pathogens, from hosts during their blood meals. More diverse hosts and larger numbers of vertebrates hosts hypothetically drive to more diverse microbiota [60]. This is probably the case for the outdoor H. marginatum and H. excavatum ticks, indeed, these tick juveniles feed, according to their tropism, on a variety of small terrestrial mammals and birds, whilst their adults can engorge on a variety of large herbivores such as goats, sheep, cattle, horses, camels [61,62]. In contrast, H. scupense, exhibits a domestic behavior, and cattle are by far the almost exclusive host for adults and juveniles in Tunisia [5,63]. Other studies found that the use of different tick species feeding on the same host don’t even have the same microbiomes, suggesting therefore, that tick microbiome is, to a certain degree, governed by species specific determinants [64]. Please refer to p13 lines 13 – 27.

P8: Mention influence on tick physiology functional role of this endosymbiont for further insights, better to rewrite

The following text was added as suggested:

Our results on symbionts of Hyalomma ticks may contribute to improve our knowledge on the interaction between tick bacterial communities and their hosts opening the way to develop subsequently applied research targeting the development of new control options against ticks and tick-borne pathogens. Bacterial endosymbionts may affect tick’s physiology and their reproductive fitness, they may also influence tick’s vectorial capacity for transmitted pathogens, and finally they may interact with the tick hosts with potential veterinary and zoonotic outcomes in particular for Francisella and Rickettsia bacteria. Please refer to p14 lines 16 – 21.

---

## [Decision Letter · Decision Letter 1]

14 Mar 2022

PONE-D-21-29206R1First insights into the microbiome of Tunisian Hyalomma ticks gained through next-generation sequencing with a special focus on H. scupensePLOS ONE

Dear Dr. Mohamed Aziz Darghouth,

Thank you for submitting your manuscript to PLOS ONE. After careful consideration, we feel that it has merit but does not fully meet PLOS ONE’s publication criteria as it currently stands. Therefore, we invite you to submit a revised version of the manuscript that addresses the points raised during the review process.

ACADEMIC EDITOR: The authors should be reply to the reviewer comments and highlight the corrections in all the manuscript

We look forward to receiving your revised manuscript.

Kind regards,

Shawky M Aboelhadid, PhD

Academic Editor

PLOS ONE

Journal Requirements:

Reviewers' comments:

Reviewer's Responses to Questions

**Comments to the Author**

1. If the authors have adequately addressed your comments raised in a previous round of review and you feel that this manuscript is now acceptable for publication, you may indicate that here to bypass the “Comments to the Author” section, enter your conflict of interest statement in the “Confidential to Editor” section, and submit your "Accept" recommendation.

Reviewer #2: All comments have been addressed

2. Is the manuscript technically sound, and do the data support the conclusions?

Reviewer #2: Yes

3. Has the statistical analysis been performed appropriately and rigorously? 

Reviewer #2: Yes

4. Have the authors made all data underlying the findings in their manuscript fully available?

Reviewer #2: No

5. Is the manuscript presented in an intelligible fashion and written in standard English?

Reviewer #2: Yes

6. Review Comments to the Author

Reviewer #2: Even though all the comments have been responded but i can't see the file with the changes in different coloured text

7. PLOS authors have the option to publish the peer review history of their article (what does this mean?). If published, this will include your full peer review and any attached files.

Reviewer #2: **Yes: **Muhammad Imran Rashid

---

## [Author Response · Author response to Decision Letter 1]

5 Apr 2022

We would like to thank the reviewers for their thoughtful comments and efforts towards improving our manuscript. In the following, we addressed comments to reviewer requirements below. In our response letter, the reviewers’ comments are numbered and colored in grey if no changes are needed and in black color if changes are required. The corresponding responses (prefaced by “Author response”) follow below, in blue color. Corresponding changes are red colored in the manuscript text on the revised file (Revised Manuscript with track changes).

1. If the authors have adequately addressed your comments raised in a previous round of review and you feel that this manuscript is now acceptable for publication, you may indicate that here to bypass the “Comments to the Author” section, enter your conflict of interest statement in the “Confidential to Editor” section, and submit your "Accept" recommendation.

Reviewer #2: All comments have been addressed

2. Is the manuscript technically sound, and do the data support the conclusions?

Reviewer #2: Yes

3. Has the statistical analysis been performed appropriately and rigorously?

Reviewer #2: Yes

4. Have the authors made all data underlying the findings in their manuscript fully available?

Reviewer #2: No

Author response:

Done, the following text was added:

"The relevant accession numbers used to access to our data are available on the following links https://www.ncbi.nlm.nih.gov/sra/PRJNA791601 (for all data)."

Please refer to page 8 lines 13 and 14

---

## [Decision Letter · Decision Letter 2]

25 Apr 2022

First insights into the microbiome of Tunisian Hyalomma ticks gained through next-generation sequencing with a special focus on H. scupense

PONE-D-21-29206R2

Dear Dr. Mohamed Aziz ,

We’re pleased to inform you that your manuscript has been judged scientifically suitable for publication and will be formally accepted for publication once it meets all outstanding technical requirements.

Kind regards,

Shawky M Aboelhadid, PhD

Academic Editor

PLOS ONE

Additional Editor Comments (optional):

Reviewers' comments:

Reviewer's Responses to Questions

**Comments to the Author**

1. If the authors have adequately addressed your comments raised in a previous round of review and you feel that this manuscript is now acceptable for publication, you may indicate that here to bypass the “Comments to the Author” section, enter your conflict of interest statement in the “Confidential to Editor” section, and submit your "Accept" recommendation.

Reviewer #2: All comments have been addressed

2. Is the manuscript technically sound, and do the data support the conclusions?

Reviewer #2: Yes

3. Has the statistical analysis been performed appropriately and rigorously? 

Reviewer #2: Yes

4. Have the authors made all data underlying the findings in their manuscript fully available?

Reviewer #2: Yes

5. Is the manuscript presented in an intelligible fashion and written in standard English?

Reviewer #2: Yes

6. Review Comments to the Author

Reviewer #2: The study was interesting and i congratulate all authors for their efforts. They have incorporated all changes.

Goodluck

7. PLOS authors have the option to publish the peer review history of their article (what does this mean?). If published, this will include your full peer review and any attached files.

Reviewer #2: **Yes: **Muhammad Imran Rashid

---

## [Editor Report · Acceptance letter]

29 Apr 2022

PONE-D-21-29206R2 

First insights into the microbiome of Tunisian *Hyalomma* ticks gained through next-generation sequencing with a special focus on *H. scupense*

Dear Dr. Darghouth:

I'm pleased to inform you that your manuscript has been deemed suitable for publication in PLOS ONE. Congratulations! Your manuscript is now with our production department. 

Kind regards, 

on behalf of

Professor Shawky M Aboelhadid 

Academic Editor

PLOS ONE